# Modeling the Spatial Formation Mechanism of Poverty-Stricken Counties in China by Using Geographical Detector

**Lei Zhou [1,2], Feng Zhen [2], Yiqing Wang [3] and Liyang Xiong [3,4,5,*]**

[1]  School of Geographic and Biologic Information, Nanjing University of Posts and Telecommunications, Nanjing 210023, China

[2]  School of Architecture and Urban Planning, Nanjing University, Nanjing 210093, China

[3]  School of Geography, Nanjing Normal University, Nanjing 210023, China

[4]  Key Laboratory of Virtual Geographic Environment, Nanjing Normal University, Ministry of Education, Nanjing 210023, China

[5]  Jiangsu Center for Collaborative Innovation in Geographical Information Resource Development and Application, Nanjing 210023, China

*  Correspondence: xiongliyang@njnu.edu.cn; Tel.: +86-1525-187-4227

**Abstract:** The poverty-stricken counties in China follow a spatial pattern of regional poverty. Examining the influential factors of this spatial pattern can provide an important reference that can guide China in its implementation of a poverty alleviation policy. By applying a geographical detector and using a sample of poverty-stricken counties in China, this study explores the spatial relationship of county distribution with spatial influential factors, including terrain relief, cultivated land quality, water resource abundance, road network density, and the locational index. These poverty-stricken counties are then classified, and the main factors that restrict their economic development are determined. The results highlight that the selected poverty-stricken counties suffer a severe condition in each of the spatial factors mentioned above. Most of these counties are classified under the location index, terrain relief, and road network density constraint types. Each of the aforementioned spatial influential factors has unique controlling mechanisms on the distribution of these poverty-stricken counties. Most of these counties are constrained by two or multiple spatial influential factors, except for some counties located in South and Central China, which are mainly constrained by a single spatial influential factor. Therefore, these single factor-constrained poverty-stricken counties warrant more attention when a developmental policy for poverty alleviation is to be implemented. The various aspects of poverty-stricken counties constrained by multiple factors must be comprehensively considered with a special focus on their development. The differentiated policies must be designed for these poverty-stricken counties on the basis of their spatial influential factors.

**Keywords:** poverty-stricken counties; spatial distribution; influential factors; geographical detector

---

## 1. Introduction

The imbalanced development of the different regions in China has attracted much attention from geographic scholars [1–3]. With the accelerating industrialization and urbanization processes in China, this imbalanced development has led to the formation of many poverty-stricken counties and the gradual emergence of a unique spatial distribution pattern that shows an apparent spatial islanding effect of regional poverty [4,5]. This specific spatial distribution pattern is shaped by various influential factors with different driving mechanisms [6,7]. Thus, the spatial formation mechanism of poverty-stricken counties in China must be explored to achieve a scientific understanding of

their natural and social backgrounds. These spatial formation mechanisms can also deepen our understanding of the spatial status quo and evolutionary process of China's county development. Given that China has implemented several poverty alleviation policies in recent years [8–10], the findings of this work can add value to strategic policy-making in China.

Many scholars have examined regional poverty in China from the perspective of spatial locations, especially in determining different regional types of poverty [4,11–14]. Wang divided China's poverty-stricken areas into two main types according to their topography [11]. Those areas classified under the first type are located in arid regions, which are represented by the Qinghai–Tibet Plateau and the Loess Plateau, while those classified under the second type are located in the southeast hilly areas and the southwest karst areas with high terrain relief and low per capita cultivated land. Cheng and Ding identified natural conditions and resources as important factors that cause and exacerbate rural poverty [12]. Zhang et al. found that rural poverty is mainly distributed in areas with poor natural conditions and inconvenient transportation [13]. Wei suggested that the poverty status of an area depends largely on its location [14]. A higher poverty rate has been observed in the western and mountainous areas of China, whereas a lower poverty rate has been recorded in the eastern and plain areas [14]. Some other scholars also found that most of the poor population in China are concentrated in areas with fragile ecological environments, poor living conditions, low productivity, and a high incidence of diseases [15–17]. In addition, the China Rural Poverty Alleviation and Development Program (2011–2020) issued by the State Council in 2011 delineated 14 contiguous poverty-stricken counties at the national level to help the government in its poverty alleviation efforts [1,4]. However, the spatial influential factors of these poverty-stricken counties are simply attributed to the natural environment, even if these factors vary significantly across areas with different driving mechanisms [4].

The multiple and complex factors, which can be mainly classified into natural and social factors, can affect the spatial formation mechanism of poverty [4]. Among these factors, the natural environment is considered the most basic natural factor that constrains the economic development of an area. Therefore, this factor must serve as an external objective condition for regional economic development. Other natural factors, including climate adaptability, water source sufficiency, and land availability, can determine the level of regional economic development in varying degrees from the perspective of natural objective conditions [18–24]. China's poverty-stricken areas greatly depend on their available land for survival, and poverty-stricken counties depend on their land for their wealth creation. Meanwhile, social factors, including the economic location and region accessibility, also have important effects on the regional population distribution and socio-economic development [25–33]. The places near main traffic routes and regional growth poles can easily obtain important developmental resources, including the land, talents, funds, and markets, whereas the places located in complex mountainous areas are often constrained by their lack of access to traffic, thereby negatively influencing their socio-economic development [30,33]. In this case, many factors with obvious regional characteristics can significantly influence regional poverty, while regional poverty appears to interact with multiple natural and social factors.

Previous studies have largely focused on identifying poverty-stricken areas, their spatial distribution, and their relationship with their natural-social background [34,35]. However, each poverty-stricken area has unique spatial formation mechanisms, and different factors can control these areas at varying levels. Therefore, the internal mechanism that drives the relationship between these poverty-stricken areas and their multiple influential factors must be quantitatively analyzed to fully comprehend the classification of regional poverty types. The geographical detector model is a quantitative statistical method for detecting spatial heterogeneity and the underlying spatial driving factors [36]. This model assumes that two variables with similar spatial distributions are statistically correlated, and it has been widely applied in geographical research [35,36]. As the most basic administrative units in China, counties can be used as statistical units in analyzing the spatial distribution of poverty [4]. Therefore, this paper analyzes the spatial constraints of regional economic development by taking counties as statistical units. The internal spatial formation mechanism of

poverty is explored by using the geographical detector model. The results are then used to classify the poverty-stricken counties and to reveal the regional formation mechanism of their spatial differentiation.

## 2. Materials and Methods

### 2.1. Materials

In 2016, the State Council of China published the List of Key Counties for National Poverty Alleviation and Development Work, which contained the names of poverty-stricken counties in different provinces [1,4]. These counties were selected as the research objects of this study (Table 1 and Figure 1a), and the relevant data were collected from the shuttle radar topography mission digital elevation model (SRTM DEM) (Figure 1b), the digital line graph (DLG) river network (Figure 1c), the national traffic map of 2016 (Figure 1d), and the 2017 China Statistical Yearbook [1,4,30,37,38].

The basic elevation data were collected from SRTM DEM (with 90 m cell size) to understand the topographical conditions of the selected counties. The data from the DLG river network were used to evaluate the water resource quality. The national traffic map of 2016 contains three types of roads, namely, highways, national roads, and railways, which are used to quantitatively determine the accessibility of an area. The data on the GDP, per capita income, and total crop area of the selected counties were collected from the 2017 China Statistical Yearbook [37].

**Table 1.** The number of poverty counties and per capita GDP of each province in 2016.

| Region | Province | Abbr. | Per Capita GDP ($10^4$ Yuan) | Poverty-Stricken Counties | |
|---|---|---|---|---|---|
| | | | | Numbers | Percentage (%) |
| Dongbei Region | Jilin | JL | 5.41 | 8 | 1.35 |
| | Heilongjiang | HLJ | 4.04 | 14 | 2.36 |
| Huabei Region | Inner Mongolia | IM | 7.42 | 31 | 5.24 |
| | Hebei | HEB | 4.29 | 39 | 6.59 |
| | Shanxi | SX | 3.53 | 35 | 5.91 |
| Huanan Region | Hainan | HN | 4.44 | 5 | *0.84* |
| | Guangxi | GX | 3.80 | 28 | 4.73 |
| Huazhong Region | Hubei | HB | 5.52 | 25 | 4.22 |
| | Hunan | HUN | 4.61 | 20 | 3.38 |
| | Henan | HEN | 4.24 | 31 | 5.24 |
| | Jiangxi | JX | 4.02 | 21 | 3.55 |
| Huadong Region | Anhui | AH | 3.93 | 19 | 3.21 |
| Xibei Region | Shaanxi | SAX | 5.05 | 50 | *8.45* |
| | Ningxia | NX | 4.72 | 8 | 1.35 |
| | Qinghai | QH | 4.38 | 15 | 2.53 |
| | Xinjiang | XJ | 4.05 | 27 | 4.56 |
| | Gansu | GS | 2.75 | 43 | *7.26* |
| Xinan Region | Chongqing | CQ | 5.82 | 14 | 2.36 |
| | Sichuan | SC | 3.98 | 36 | 6.08 |
| | Guizhou | GZ | 3.32 | 50 | 8.45 |
| | Yunnan | YN | 3.14 | 73 | *12.33* |
| Total | | | | 592 | 100.00 |

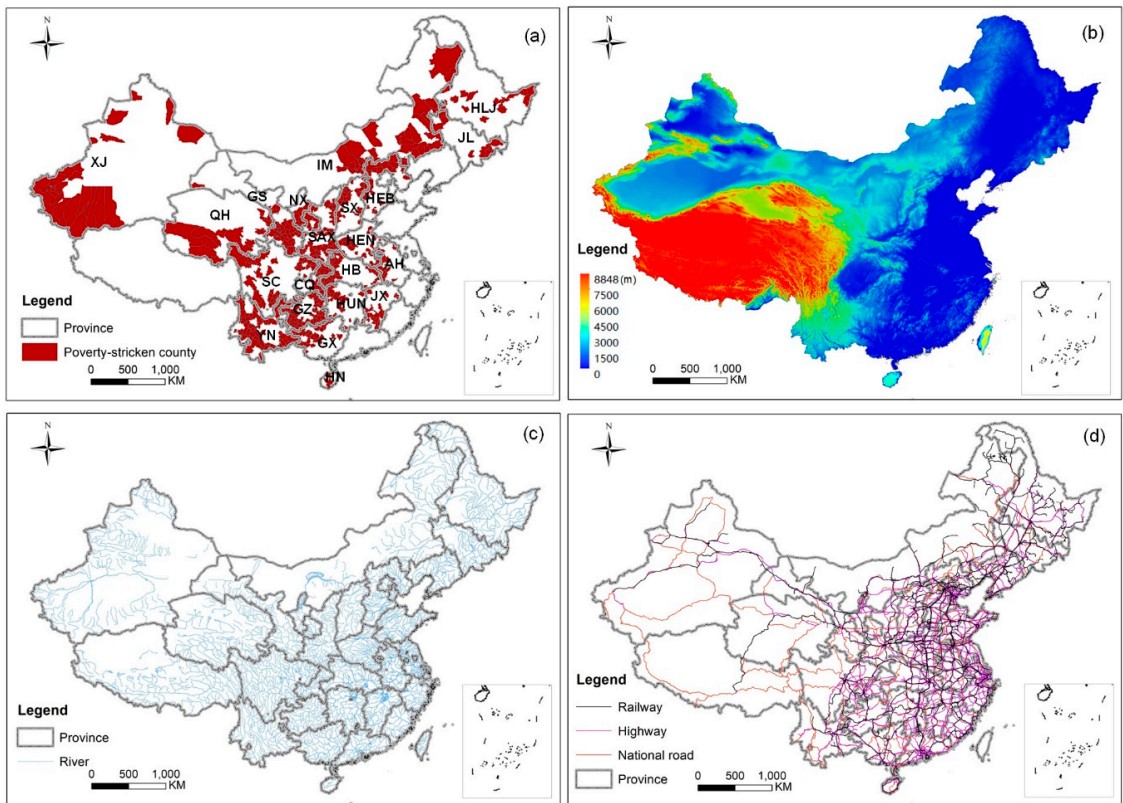

**Figure 1.** Poverty-stricken counties in China and the data used in this study. (**a**) Poverty-stricken counties, (**b**) shuttle radar topography mission digital elevation model (SRTM DEM), (**c**) the river network, and (**d**) the transportation network.

## 2.2. Calculation of Spatial Influential Factors

As mentioned in the Introduction, the economic development of the counties is affected by a combination of social and natural factors. Therefore, the spatial influential factors should cover both the social and natural aspects. An index system was developed in this work to quantitatively express the spatial factors that have influenced the economic development of a county. This system employs the factors of terrain relief ($X_1$), cultivated land quality ($X_2$), and water resource abundance ($X_3$) as natural influential constraints and the factors of road network density ($X_4$) and location index ($X_5$) as social influential constraints [4]. The additional information on these factors and their calculations are presented in Table 2 [3,4,39,40]. Among these factors, terrain relief can demonstrate the topographic environment of a county [39], cultivated land quality and water resource abundance can express its natural resource endowments [4], and road network density and location index can be used to represent its accessibility and economic location [33,40], respectively. These five spatial influential factors are expected to highlight the spatial disparities of poverty-stricken counties in China. With the use of the calculation methods presented in Table 2, the selected poverty-stricken counties were classified as very poor, relatively poor, fair, relatively good, and good based on the five aforementioned factors with the natural break method in ArcGIS software. This natural break classification method can help maximize the differences between the different categories with the natural turning point. Thus, the differences between the different categories of poverty-stricken counties can be determined.

**Table 2.** The spatial influential factors for spatial restrictions in poverty-stricken counties.

| Types | Spatial Constraints | Calculation Formula | Notes of the Formula |
|---|---|---|---|
| Natural factors | Topographic relief ($X_1$) | $X_1 = H_{\max} - H_{\min}$ | $H_{max}$ and $H_{min}$ are the maximum and minimum elevation values. |
| | Cultivated land quality ($X_2$) | $X_2 = \frac{P - P_{\min}}{P_{\max} - P_{\min}} \times n$ | $P$ is the county's population, $P_{max}$ and $P_{min}$ are the maximum and minimum population of all counties in China, $n$ is the total crop area of the county. |
| | Water abundance ($X_3$) | $X_3 = \frac{Len}{S}$ | $Len$ is the length of water passing through the county, and $S$ is the area of the county. |
| Social factors | Road network density ($X_4$) | $X_4 = C_1 \times 0.3 + C_2 \times 0.4 + C_3 \times 0.3$ | $C_1$, $C_2$, and $C_3$ are the ratios of the length of national road, highway and railway, and the county area. |
| | Location index ($X_5$) | $X_5 = a \times b$ $a = \begin{cases} 1, L \geq 360 \\ 1.5, 180 \leq L < 360 \\ 2, 0 \leq L < 180 \end{cases}$ $b = \frac{\sqrt{e \times p}}{\sqrt{e_0 \times p_0}}$ | $L$ is the distance between the county and its provincial capital. $e$ and $p$ are the GDP and population of the county; $e_0$, and $p_0$ are the average values of GDP and population of all counties in the province. |

*2.3. Geographical Detector Model*

The geographical detector model is a statistical method used to detect the spatial relationship among different spatial variables. Although the correlation is not causation, this geographical detector model assumes that if an independent variable has a significant influence on another dependent variable, then the spatial distribution of these variables should show some similarities [36]. In this study, the geographical detector model was used to reveal the spatial formation mechanism of poverty-stricken counties. The basic idea is that the influential factors that affect the developmental level of a county can be calculated and distributed over space, while the economic levels of poverty-stricken counties can also be distributed over space. Therefore, if a significant spatial consistency is observed between a certain spatial influential factor and the economic development level of a county, then this spatial influential factor profoundly determines the economic development level of this county. In the geographical detector model, the dependent variable ($Y$) denotes the spatial distribution of a county's economic development level, while the spatial distribution of influential factors ($X_i$, $i$ = 1, 2, 3, 4, 5) is treated as the independent variable. The determinant value ($q$) from the independent variables to the dependent variable can be calculated as follows by using this model [36]:

$$q = 1 - \frac{\sum\limits_{h=1}^{L} N_h \sigma_h^2}{N \sigma^2} \tag{1}$$

where $h$ is a certain subregion (county level) of variable $Y$ or $X$ ($h$ = 1, ... , $L$), L is the number of partitions, $N$ is the number of counties in the entire region (province level), $N_h$ is the number of counties in a subregion $h$, $\sigma^2$ denotes the variance in the economic development level of counties within the entire region, and $\sigma^{2h}$ denotes the variance in the economic development level of counties within subregion $h$. The value of $q$ has a range of [0,1]. The geographical detector model explores the spatial relationship between the spatial pattern of poverty-stricken counties and their influential factors. Thus, an observation unit for factor calculation and a statistical unit for spatial relationship detection are needed. In this paper, the county boundary is regarded as the observation unit, while the province boundary is regarded as the statistical unit in the geographical detector. A larger $q$ indicates that the spatial influential factor $X$ has a greater impact on the spatial distribution of county economic development level $Y$, and vice versa. Moreover, when $q$ is 1, the spatial influential factor $X$ completely controls the spatial distribution of county economic development level $Y$, but when $q$ is 0, the county economic development level is randomly distributed and is not affected by the spatial influential factor

*X*. In addition, the economic development of a county is often affected by multiple spatial influential factors that often demonstrate a complex inner relationship. Therefore, with the geographical detector model, a combined analysis that uses different combinations of spatial influential factors should be performed to understand the economic development of poverty-stricken counties. Three types of combinations, namely, single-, double-, and multiple-factor combinations, are employed based on the number of core dominant spatial influential factors that drive a county's economic development.

## 3. Results

### 3.1. Spatial Distribution of Influential Factors

#### 3.1.1. Terrain Relief

Figure 2 presents the spatial distribution of terrain relief over different counties. The five levels of terrain relief conditions are classified for the poverty-stricken and all counties. These counties show great variations in their terrain relief conditions. The counties with very poor and relatively poor topographic conditions account for 45.1% and 25.7% of all poverty-stricken counties, respectively, while counties with relatively good and very good topography conditions account for only 9.0% and 10.3%, respectively. In other words, most poverty-stricken counties in China have a poor topographical environment. Meanwhile, only 44.5% of all countries in China have very poor and relatively poor topographic conditions. However, this percentage is far below than that recorded in the poverty-stricken counties (i.e., 70.8%). According to the comparison of the five levels of topographical conditions of all counties and poverty-stricken counties in China, the proportions of poverty-stricken counties with very poor, relatively poor, fair, relatively good, and very good topographic conditions increased by +21%, +5.3%, −8%, −11.2%, and −7.1%, respectively. In sum, the poverty-stricken counties have poorer topographical conditions compared with the other counties in China.

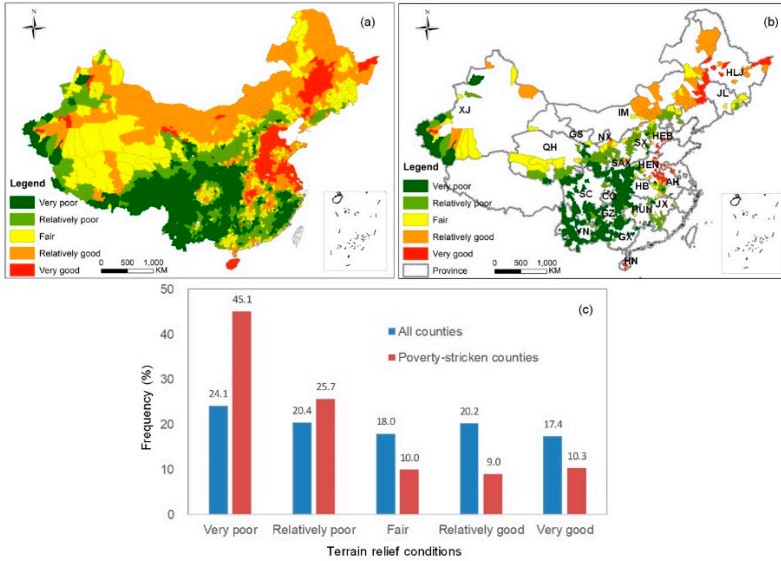

**Figure 2.** The spatial distribution characteristics of terrain relief. (**a**) All counties; (**b**) poverty-stricken counties; and (**c**) a comparison of grading proportions between the all-counties and poverty-stricken counties samples.

Table 3 shows the topographical conditions of poverty-stricken counties by province. Among these provinces, 13 have more than 50% counties with very poor and relatively poor topographical conditions. These provinces include Guangxi (100%), Guizhou (100%), Sichuan (100%), Yunnan (100%), Gansu (95.3%), Hunan (95%), Chongqing (92.9%), Jiangxi (90.5%), Hubei (80%), Shaanxi (80%), Shanxi (77.2%), Qinghai (73.3%), and Ningxia (62.5%), most of which are located in the central and western

regions of China. The poverty-stricken counties in Xinjiang, Qinghai, Hebei, Henan, and Hubei show substantial differences in their terrain conditions, while counties with good topographical conditions are mainly located in Inner Mongolia, Heilongjiang, and Xinjiang.

**Table 3.** Overview of the topographic relief level of poverty-stricken counties (Unit: %, the bold number means the sum of very poor and relatively poor frequencies larger than 50%).

| Province | Very Poor | Relatively Poor | Fair | Relatively Good | Very Good |
|---|---|---|---|---|---|
| Anhui | 10.5 | 21.1 | 5.3 | 10.5 | 52.6 |
| Gansu | **37.2** | **58.1** | 4.7 | 0.0 | 0.0 |
| Guangxi | **85.7** | **14.3** | 0.0 | 0.0 | 0.0 |
| Guizhou | **82.0** | **18.0** | 0.0 | 0.0 | 0.0 |
| Hainan | 0.0 | 0.0 | 0.0 | 0.0 | 100.0 |
| Hebei | 0.0 | 28.2 | 20.5 | 15.4 | 35.9 |
| Henan | 12.9 | 12.9 | 16.1 | 6.5 | 51.6 |
| Heilongjiang | 0.0 | 0.0 | 0.0 | 42.9 | 57.1 |
| Hubei | **64.0** | **16.0** | 20.0 | 0.0 | 0.0 |
| Hunan | **55.0** | **40.0** | 5.0 | 0.0 | 0.0 |
| Jilin | 0.0 | 12.5 | 50.0 | 0.0 | 37.5 |
| Jiangxi | **9.5** | **81.0** | 4.8 | 4.8 | 0.0 |
| Xinjiang | 25.9 | 14.8 | 25.9 | 25.9 | 7.4 |
| Ningxia | **12.5** | **50.0** | 25.0 | 12.5 | 0.0 |
| Qinghai | **33.3** | **40.0** | 26.7 | 0.0 | 0.0 |
| Shanxi | **8.6** | **68.6** | 20.0 | 2.9 | 0.0 |
| Shaanxi | **48.0** | **32.0** | 8.0 | 10.0 | 2.0 |
| Sichuan | **83.3** | **16.7** | 0.0 | 0.0 | 0.0 |
| Inner Mongolia | 0.0 | 0.0 | 22.6 | 71.0 | 6.5 |
| Yunnan | **93.2** | **6.8** | 0.0 | 0.0 | 0.0 |
| Chongqing | **92.9** | **0.0** | 7.1 | 0.0 | 0.0 |
| National | 45.1 | 25.7 | 10.0 | 9.0 | 10.3 |

### 3.1.2. Cultivated Land Quality

Figure 3 shows small differences in the cultivated land quality of poverty-stricken counties. The counties with very poor and relatively poor cultivated land quality account for 48.8% of the poverty-stricken counties sample, while the counties with very good cultivated land quality account for only 14.4%. In sum, the quality of the cultivated land in poverty-stricken counties is relatively poor. According to the comparison of the five levels of cultivated land qualities of all counties and poverty-stricken counties in China, the proportions of poverty-stricken counties with very poor, relatively poor, fair, relatively good, and very good cultivated land qualities increased by +4%, +5.6%, −1.1%, −2%, and −6.4%, respectively. No significant difference was observed in these percentages between the total counties and poverty-stricken counties samples.

Table 4 shows that seven provinces have more than 50% counties with very poor and relatively poor cultivated land quality. These provinces are Yunnan (97.2%), Shanxi (88.6%), Hainan (80%), Qinghai (73.3%), Shaanxi (68%), Xinjiang (66.6%), and Sichuan (63.9%). Meanwhile, Qinghai, Shaanxi, and Anhui show significant differences in the cultivated land quality of their poverty-stricken counties, whereas the poverty-stricken counties in Heilongjiang and Chongqing have good cultivated land quality.

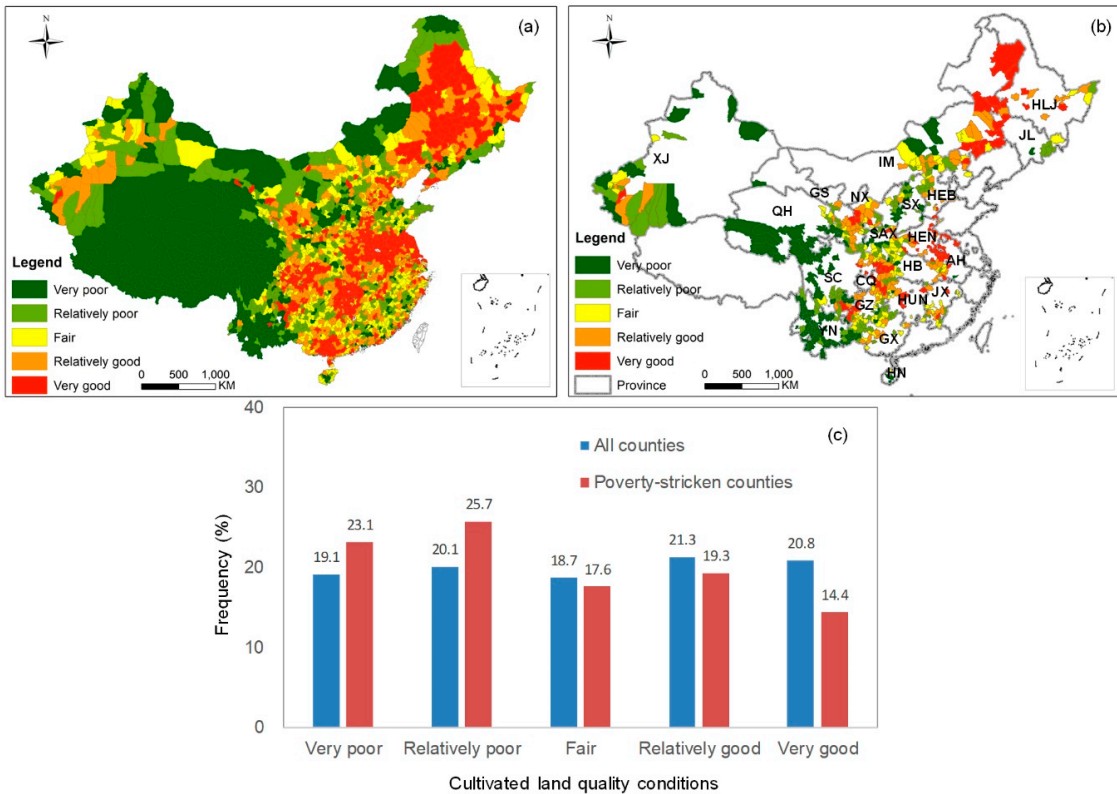

**Figure 3.** The spatial distribution characteristics of cultivated land quality. (**a**) All counties; (**b**) poverty-stricken counties; and (**c**) a comparison of grading proportions between the all-counties and poverty-stricken counties samples.

**Table 4.** Overview of the cultivated land quality of the poverty-stricken counties (Unit: %, the bold number means the sum of very poor and relatively poor frequencies larger than 50%).

| Province | Very Poor | Relatively Poor | Fair | Relatively Good | Very Good |
|---|---|---|---|---|---|
| Anhui | 5.3 | 0.0 | 10.5 | 15.8 | 68.4 |
| Gansu | 14.0 | 30.2 | 14.0 | 37.2 | 4.7 |
| Guangxi | 0.0 | 39.3 | 32.1 | 28.6 | 0.0 |
| Guizhou | 8.0 | 30.0 | 32.0 | 18.0 | 12.0 |
| Hainan | **60.0** | **20.0** | 20.0 | 0.0 | 0.0 |
| Hebei | 2.6 | 25.6 | 28.2 | 35.9 | 7.7 |
| Henan | 0.0 | 6.5 | 6.5 | 29.0 | 58.1 |
| Heilongjiang | 0.0 | 7.1 | 14.3 | 42.9 | 35.7 |
| Hubei | 4.0 | 36.0 | 32.0 | 28.0 | 0.0 |
| Hunan | 5.0 | 20.0 | 25.0 | 25.0 | 25.0 |
| Jilin | 12.5 | 37.5 | 12.5 | 12.5 | 25.0 |
| Jiangxi | 0.0 | 33.3 | 33.3 | 23.8 | 9.5 |
| Xinjiang | **33.3** | **33.3** | 14.8 | 14.8 | 3.7 |
| Ningxia | 12.5 | 12.5 | 25.0 | 25.0 | 25.0 |
| Qinghai | **53.3** | **20.0** | 20.0 | 6.7 | 0.0 |
| Shanxi | **40.0** | **48.6** | 8.6 | 2.9 | 0.0 |
| Shaanxi | **24.0** | **44.0** | 20.0 | 8.0 | 4.0 |
| Sichuan | **36.1** | **27.8** | 8.3 | 19.4 | 8.3 |
| Inner Mongolia | 22.7 | 19.8 | 19.0 | 12.6 | 25.9 |
| Yunnan | **80.8** | **16.4** | 0.0 | 1.4 | 1.4 |
| Chongqing | 0.0 | 0.0 | 0.0 | 28.6 | 71.4 |
| National | 23.1 | 25.7 | 17.6 | 19.3 | 14.4 |

### 3.1.3. Water Resource Abundance

Considerable variations can be found in the water resource abundance conditions of the poverty-stricken counties (Figure 4). The counties with fair water resource abundance conditions constitute the majority of the poverty-stricken counties sample (32.8%), while those with very poor and very good water resource abundance conditions account for 9.1% and 10.3%, respectively. In general, the water resource abundance conditions of these counties are not as bad as their terrain relief conditions. According to the comparison of the five levels of water conditions of all counties and poverty-stricken counties in China, the proportions of poverty-stricken counties with very poor, relatively poor, fair, relatively good, and very good water conditions increased by −0.5%, +2.5%, +9.1%, −0.1%, and −11%, respectively. Compared with the all-counties sample, relatively fewer counties in the poverty-stricken counties sample have relatively good and very good levels of water abundance conditions, suggesting the water abundance conditions of these poverty-stricken counties are below the average national level.

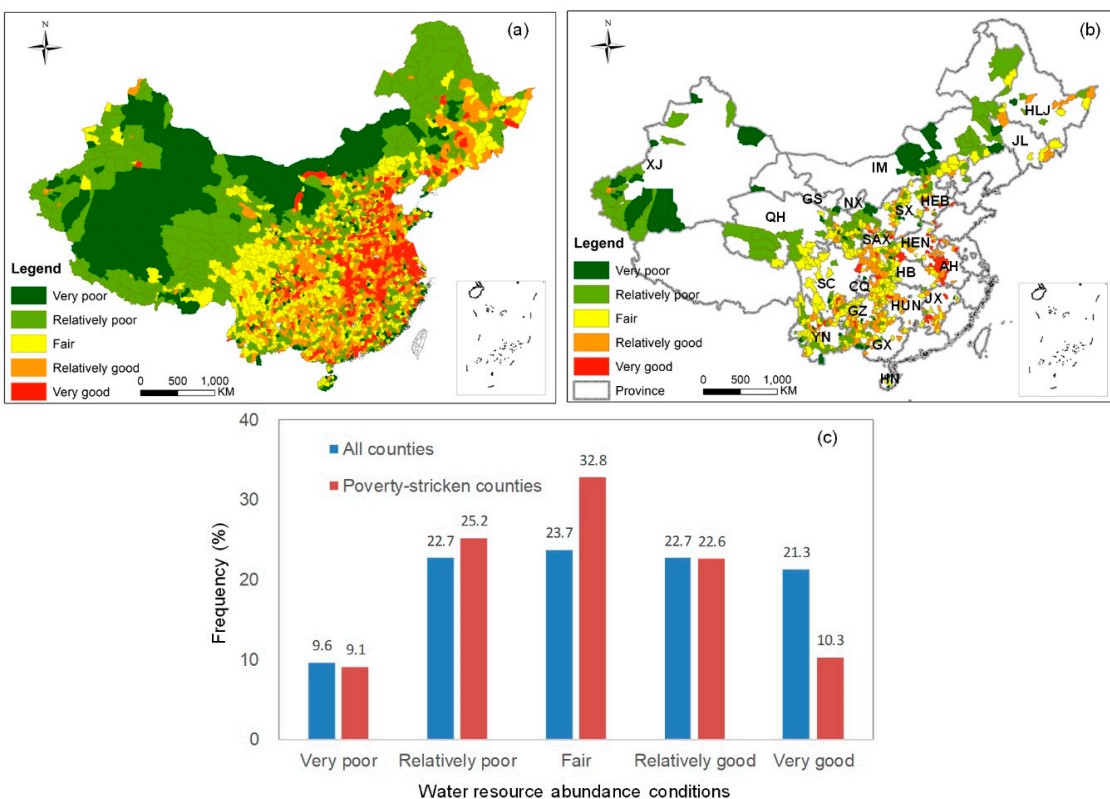

**Figure 4.** The spatial distribution characteristics of water resource abundance. (**a**) All counties; (**b**) poverty-stricken counties; and (**c**) a comparison of grading proportions between the all-counties and poverty-stricken counties samples.

Table 5 shows that six provinces have more than 50% of the counties with very poor and relatively poor water conditions. These provinces are Xinjiang (96.3%), Inner Mongolia (87.1%), Ningxia (75%), Qinghai (66.7%), Hainan (60%), and Gansu (51.2%). In addition, Gansu, Guizhou, Shanxi, and Shaanxi show major differences in the water abundance conditions of their poverty-stricken counties, whereas the poverty-stricken counties in Anhui, Jiangxi, and Sichuan have fair water abundance conditions.

**Table 5.** Overview of the water abundance level of the poverty-stricken counties (Unit: %, the bold number means the sum of very poor and relatively poor frequencies larger than 50%).

| Province | Very Poor | Relatively Poor | Fair | Relatively Good | Very Good |
|---|---|---|---|---|---|
| Anhui | 0.0 | 0.0 | 0.0 | 47.4 | 52.6 |
| Gansu | **9.3** | **41.9** | 32.6 | 11.6 | 4.7 |
| Guangxi | 10.7 | 25.0 | 28.6 | 35.7 | 0.0 |
| Guizhou | 4.0 | 10.0 | 40.0 | 42.0 | 4.0 |
| Hainan | **40.0** | **20.0** | 40.0 | 0.0 | 0.0 |
| Hebei | 5.1 | 25.6 | 28.2 | 15.4 | 25.6 |
| Henan | 0.0 | 9.7 | 29.0 | 19.4 | 41.9 |
| Heilongjiang | 7.1 | 35.7 | 21.4 | 35.7 | 0.0 |
| Hubei | 0.0 | 24.0 | 52.0 | 16.0 | 8.0 |
| Hunan | 0.0 | 0.0 | 55.0 | 35.0 | 10.0 |
| Jilin | 12.5 | 0.0 | 37.5 | 50.0 | 0.0 |
| Jiangxi | 4.8 | 4.8 | 23.8 | 42.9 | 23.8 |
| Xinjiang | **44.4** | **51.9** | 0.0 | 3.7 | 0.0 |
| Ningxia | **25.0** | **50.0** | 25.0 | 0.0 | 0.0 |
| Qinghai | **6.7** | **60.0** | 33.3 | 0.0 | 0.0 |
| Shanxi | 8.6 | 34.3 | 42.9 | 11.4 | 2.9 |
| Shaanxi | 4.0 | 20.0 | 34.0 | 30.0 | 12.0 |
| Sichuan | 5.6 | 13.9 | 47.2 | 22.2 | 11.1 |
| Inner Mongolia | **38.7** | **48.4** | 12.9 | 0.0 | 0.0 |
| Yunnan | 5.5 | 31.5 | 39.7 | 19.2 | 4.1 |
| Chongqing | 0.0 | 7.1 | 42.9 | 42.9 | 7.1 |
| National | 9.1 | 25.2 | 32.8 | 22.6 | 10.3 |

### 3.1.4. Road Network Density

Figure 5 shows significant variations in the road network density of the poverty-stricken counties. The poverty-stricken counties with very poor and relatively poor road network conditions account for 24.2% and 34% of the poverty-stricken counties sample, and only 1.20% and 14.20% of the counties in this sample have very good and relatively good road network conditions, respectively. According to the comparison of the five levels of road conditions of all the counties and poverty-stricken counties in China, the proportions of poverty-stricken counties with very poor, relatively poor, fair, relatively good, and very good road conditions increased by +9.7%, +10.2%, +3.3%, −10.3%, and −12.8%, respectively. In sum, the road network density conditions of poverty-stricken counties in China are below the average national level.

Table 6 shows that 13 provinces have more than 50% of poverty-stricken counties with very poor and relatively poor road network density conditions. These provinces are Xinjiang (92.5%), Guangxi (82.1%), Heilongjiang (78.6%), Sichuan (77.7%), Inner Mongolia (77.4%), Jilin (75%), Ningxia (75%), Qinghai (73.3%), Yunnan (72.6%), Gansu (65.1%), Guizhou (58%), Hubei (56%), and Hunan (55%). Jiangxi, Hubei, and Chongqing show large differences in the road network density conditions of their poverty-stricken counties. Compared with the all-counties sample, relatively fewer counties in the poverty-stricken counties sample have relatively good and very good road network density conditions, thereby highlighting the backward construction of road networks in China's poverty-stricken counties.

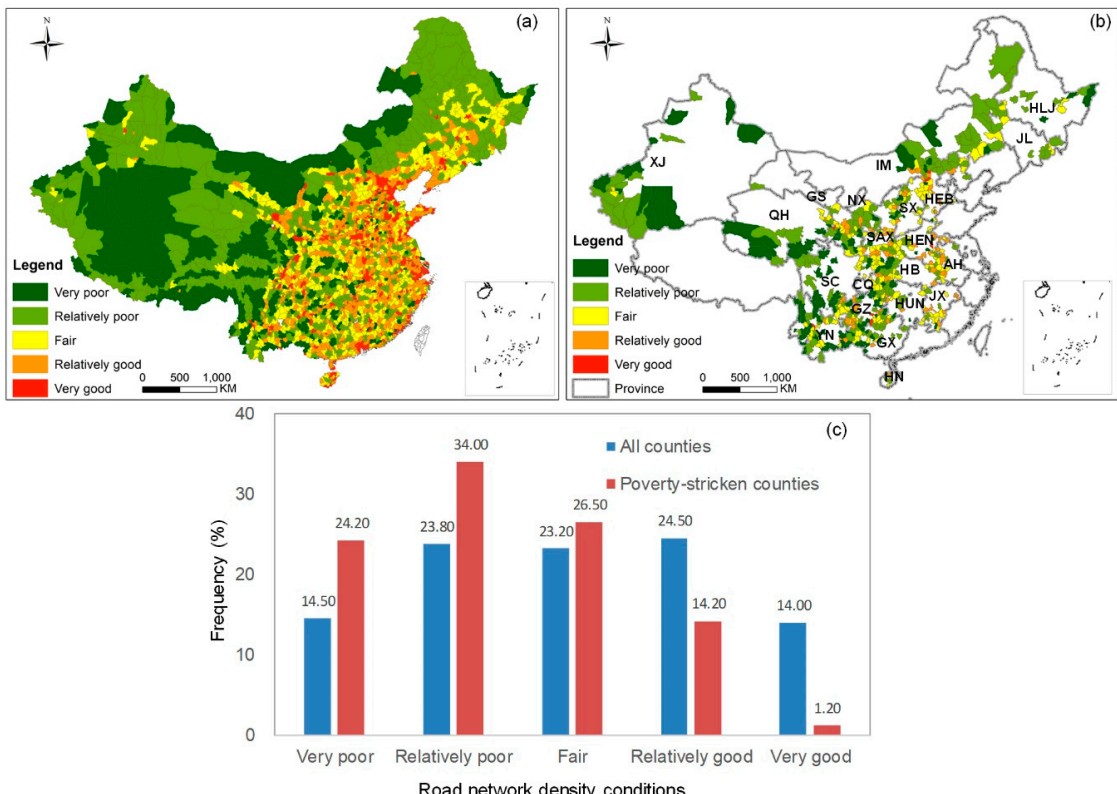

**Figure 5.** The spatial distribution characteristics of road network density. (**a**) All counties; (**b**) poverty-stricken counties; and (**c**) a comparison of grading proportions between the all-counties and poverty-stricken counties samples.

**Table 6.** Overview of the road network density level of poverty-stricken counties (Unit: %, the bold number means the sum of very poor and relatively poor frequencies larger than 50%).

| Province | Very Poor | Relatively Poor | Fair | Relatively Good | Very Good |
|---|---|---|---|---|---|
| Anhui | 0.0 | 31.6 | 36.8 | 31.6 | 0.0 |
| Gansu | **27.9** | **37.2** | 20.9 | 11.6 | 2.3 |
| Guangxi | **35.7** | **46.4** | 10.7 | 7.1 | 0.0 |
| Guizhou | **30.0** | **28.0** | 22.0 | 16.0 | 4.0 |
| Hainan | 0.0 | 40.0 | 20.0 | 40.0 | 0.0 |
| Hebei | 10.3 | 25.6 | 33.3 | 25.6 | 5.1 |
| Henan | 9.7 | 22.6 | 35.5 | 29.0 | 3.2 |
| Heilongjiang | **28.6** | **50.0** | 21.4 | 0.0 | 0.0 |
| Hubei | **16.0** | **40.0** | 32.0 | 12.0 | 0.0 |
| Hunan | **15.0** | **40.0** | 30.0 | 15.0 | 0.0 |
| Jilin | 0.0 | 75.0 | 25.0 | 0.0 | 0.0 |
| Jiangxi | 4.8 | 19.0 | 47.6 | 28.6 | 0.0 |
| Xinjiang | **48.1** | **44.4** | 7.4 | 0.0 | 0.0 |
| Ningxia | **0.0** | **75.0** | 12.5 | 12.5 | 0.0 |
| Qinghai | **40.0** | **33.3** | 20.0 | 6.7 | 0.0 |
| Shanxi | 14.3 | 31.4 | 42.9 | 11.4 | 0.0 |
| Shaanxi | 10.0 | 20.0 | 38.0 | 32.0 | 0.0 |
| Sichuan | **44.4** | **33.3** | 19.4 | 2.8 | 0.0 |
| Inner Mongolia | **16.1** | **61.3** | 16.1 | 6.5 | 0.0 |
| Yunnan | **47.9** | **24.7** | 20.5 | 5.5 | 1.4 |
| Chongqing | **14.3** | **35.7** | 42.9 | 7.1 | 0.0 |
| National | 24.2 | 34.0 | 26.5 | 14.2 | 1.2 |

### 3.1.5. Location Index

The spatial distribution of the location index is shown in Figure 6. The poverty-stricken counties show substantial variations in their location indices. The poverty-stricken counties with very poor and relatively poor location conditions account for 31.3% and 22.1% of the poverty-stricken counties sample, while only 9.1% and 14.5% of the counties in this sample have very good and relatively good location conditions, respectively. In sum, the location conditions of most poverty-stricken counties in China are relatively poor. The comparison of the five levels of location conditions of all counties and poverty-stricken counties in China indicates that the proportions of poverty-stricken counties with very poor, relatively poor, fair, relatively good, and very good location conditions increased by +10%, +2%, +1.4%, −5.3% and −8%, respectively. Therefore, the location conditions of the counties included in the poverty-stricken counties sample are worse than those of the counties included in the all counties sample.

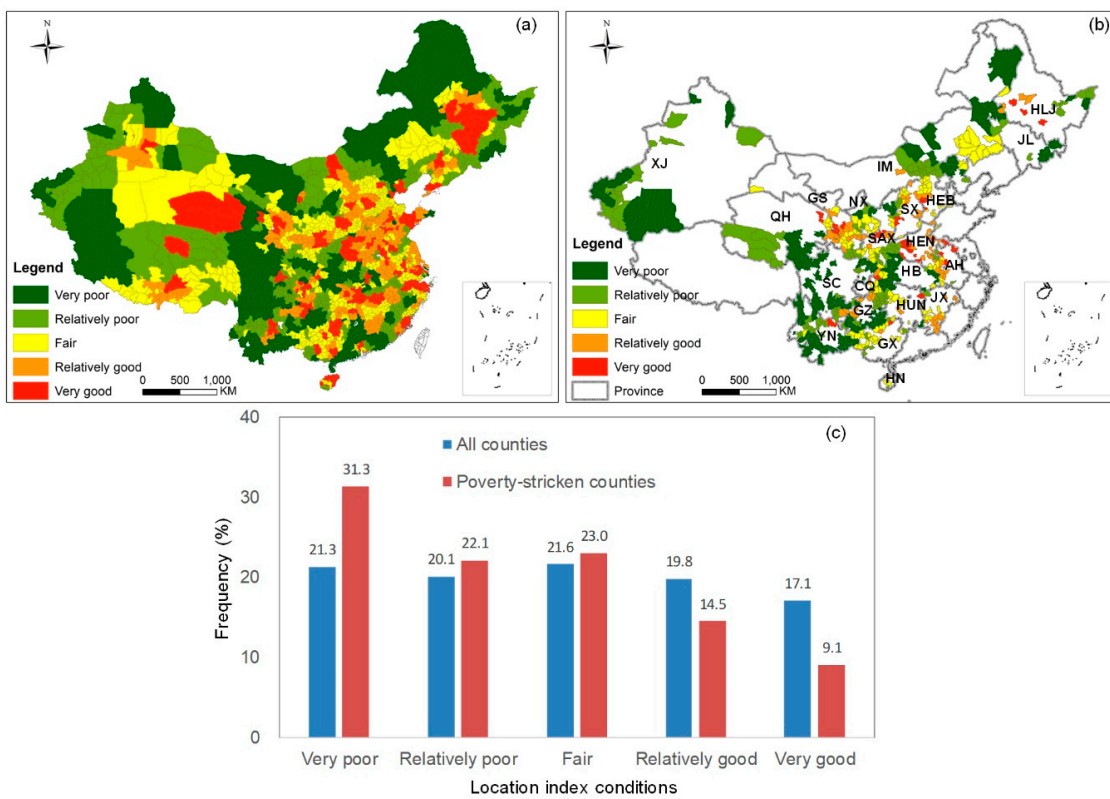

**Figure 6.** The spatial distribution characteristics of location index. (**a**) All counties; (**b**) poverty-stricken counties; and (**c**) a comparison of grading proportions between the all-counties and poverty-stricken counties samples.

Table 7 shows that nine provinces have more than 50% poverty-stricken counties with very poor and relatively poor location conditions. These provinces are Sichuan (100%), Jilin (100%), Xinjiang (100%), Hubei (96%), Yunnan (95.9%), Guizhou (82%), Ningxia (75%), Qinghai (53.4%), and Inner Mongolia (58.1%). Meanwhile, Hebei, Shaanxi, and Chongqing show large differences in the location conditions of their poverty-stricken counties, whereas poverty-stricken counties in Henan and Anhui have good location conditions.

**Table 7.** Overview of the location index level of poverty-stricken counties (Unit: %, the bold number means the sum of very poor and relatively poor frequencies larger than 50%).

| Province | Very Poor | Relatively Poor | Fair | Relatively Good | Very Good |
|---|---|---|---|---|---|
| Anhui | 0.0 | 21.1 | 15.8 | 31.6 | 31.6 |
| Gansu | 0.0 | 11.6 | 37.2 | 32.6 | 18.6 |
| Guangxi | 0.0 | 17.9 | 75.0 | 3.6 | 3.6 |
| Guizhou | **60.0** | **22.0** | 0.0 | 18.0 | 0.0 |
| Hainan | 0.0 | 20.0 | 40.0 | 0.0 | 40.0 |
| Hebei | 12.8 | 20.5 | 43.6 | 12.8 | 10.3 |
| Henan | 0.0 | 3.2 | 19.4 | 41.9 | 35.5 |
| Heilongjiang | 14.3 | 35.7 | 7.1 | 21.4 | 21.4 |
| Hubei | **92.0** | **4.0** | 4.0 | 0.0 | 0.0 |
| Hunan | 15.0 | 30.0 | 40.0 | 10.0 | 5.0 |
| Jilin | **62.5** | **37.5** | 0.0 | 0.0 | 0.0 |
| Jiangxi | 0.0 | 4.8 | 38.1 | 57.1 | 0.0 |
| Xinjiang | **48.1** | **51.9** | 0.0 | 0.0 | 0.0 |
| Ningxia | **12.5** | **62.5** | 25.0 | 0.0 | 0.0 |
| Qinghai | **26.7** | **26.7** | 33.3 | 0.0 | 13.3 |
| Shanxi | 0.0 | 8.6 | 48.6 | 25.7 | 17.1 |
| Shaanxi | 16.0 | 32.0 | 20.0 | 18.0 | 14.0 |
| Sichuan | **83.3** | **16.7** | 0.0 | 0.0 | 0.0 |
| Inner Mongolia | **32.3** | **25.8** | 38.7 | 3.2 | 0.0 |
| Yunnan | **67.1** | **28.8** | 1.4 | 0.0 | 2.7 |
| Chongqing | 14.3 | 21.4 | 42.9 | 14.3 | 7.1 |
| National | 31.3 | 22.1 | 23.0 | 14.5 | 9.1 |

*3.2. Geographical Detection Results of Spatial Influential Factors*

To further analyze the impact mechanisms of the five spatial influential factors on the spatial differentiation of poverty-stricken counties, the geographical detector model was used to examine these factors and to calculate the determinant value $q$. If the $q$ value of a poverty-stricken county is greater than the mean $q$ value of all poverty-stricken counties within the same province, then the spatial influential factors significantly influence the economic development of all poverty-stricken counties in the same province. The determinant value $q$ for each province is listed in Table 8. The mean $q$ values for terrain relief, cultivated land quality, water abundance, road network density, and location index are 0.145, 0.270, 0.161, 0.198, and 0.195, respectively. Given its highest $q$ value, cultivated land quality is the most significant spatial influential factor that restricts the economic development of poverty-stricken counties in China. This finding is consistent with the fact that these counties are mostly inhabited by rural residents and that their economic development greatly depends on the condition of cultivated land quality. The road network density (0.198) and the location index (0.195) also have relatively large $q$ values, thereby highlighting the importance of contemporary traffic conditions and the economic location in the economic development of a county and underscoring the feasibility of alleviating poverty by improving regional traffic and location conditions.

**Table 8.** *q* value of each spatial constraint.

| Province | Terrain Relief | Cultivated Land Quality | Water Abundance | Road Network Density | Location Index |
|---|---|---|---|---|---|
| Hebei | 0.294 | 0.208 | 0.226 | 0.057 | 0.416 |
| Shanxi | 0.117 | 0.055 | 0.107 | 0.079 | 0.162 |
| Jilin | 0.120 | 0.835 | 0.017 | 0.204 | 0.085 |
| Heilongjiang | 0.023 | 0.636 | 0.067 | 0.242 | 0.482 |
| Anhui | 0.519 | 0.602 | 0.011 | 0.155 | 0.326 |
| Jiangxi | 0.257 | 0.144 | 0.393 | 0.192 | 0.043 |
| Henan | 0.363 | 0.496 | 0.252 | 0.198 | 0.325 |
| Hubei | 0.181 | 0.225 | 0.062 | 0.031 | 0.014 |
| Hunan | 0.099 | 0.057 | 0.254 | 0.103 | 0.222 |
| Chongqing | 0.001 | 0.053 | 0.643 | 0.735 | 0.647 |
| Sichuan | 0.009 | 0.224 | 0.095 | 0.188 | 0.001 |
| Guizhou | 0.000 | 0.105 | 0.027 | 0.148 | 0.087 |
| Yunnan | 0.056 | 0.092 | 0.044 | 0.061 | 0.026 |
| Shaanxi | 0.198 | 0.093 | 0.058 | 0.100 | 0.070 |
| Gansu | 0.020 | 0.051 | 0.067 | 0.023 | 0.061 |
| Qinghai | 0.153 | 0.020 | 0.061 | 0.236 | 0.290 |
| Inner Mongolia | 0.015 | 0.569 | 0.145 | 0.154 | 0.072 |
| Ningxia | 0.313 | 0.455 | 0.465 | 0.970 | 0.176 |
| Guangxi | 0.091 | 0.024 | 0.190 | 0.007 | 0.255 |
| Xinjiang | 0.079 | 0.461 | 0.022 | 0.068 | 0.143 |
| Mean | 0.145 | 0.270 | 0.161 | 0.198 | 0.195 |

## 4. Discussion

### 4.1. Spatial Formation Types of Poverty-Stricken Counties

On the basis of the results of the geographical detection model, the poverty-stricken counties are classified into several types based on their spatial influential factors. These types are the terrain condition constraint, cultivated land resource constraint, water abundance constraint, traffic condition constraint, and the location index constraint types. The spatial formation mechanism of each of these types is explored as follows.

#### 4.1.1. Terrain Condition Constraint

The results of the geographical detection model identify eight provinces whose poverty-stricken counties are affected by their terrain conditions. These provinces are Anhui (0.519), Henan (0.363), Ningxia (0.313), Hebei (0.294), Jiangxi (0.257), Shaanxi (0.198), Hubei (0.181), and Qinghai (0.153) (Figure 7a). As an important natural factor, terrain relief not only represents the advantages and disadvantages of a county's topography and geomorphology, but also affects the county's agricultural production structure and mode. The terrain relief also has an important influence on the construction of transportation facilities in a specific county and the economic linkage among different locations. Generally, agricultural production is more suitable and traffic facilities are easier to build in counties with a low terrain relief. These counties also have close economic exchanges with central cities, thereby facilitating their rapid economic development. The aforementioned provinces are concentrated in the central and western regions of China, where the terrain conditions vary significantly. The economy of a county can be significantly constrained by a poor terrain condition, thereby explaining the poverty in the aforementioned provinces.

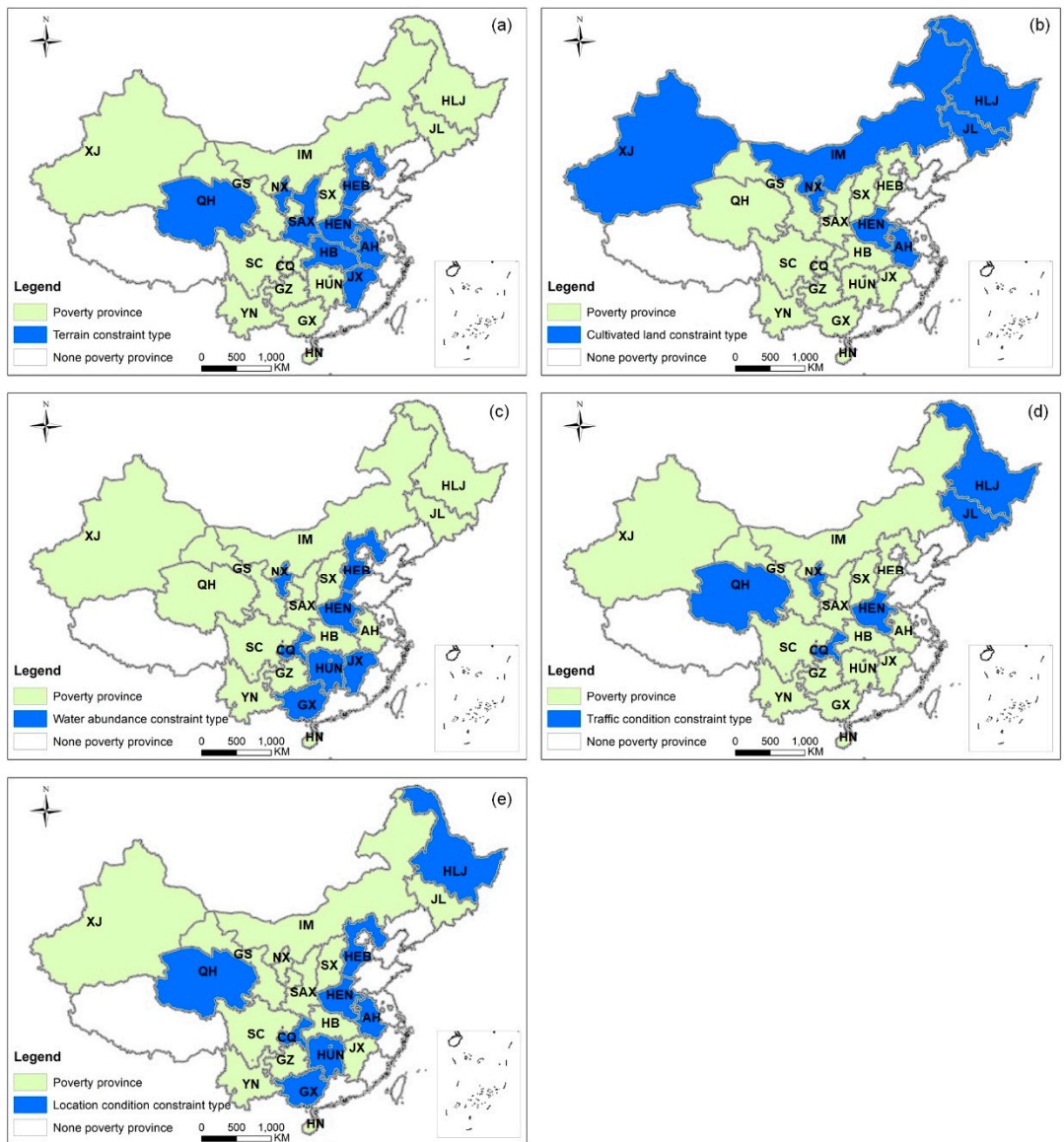

**Figure 7.** The restrictions on the development of state-level poverty-stricken counties. (**a**) Terrain condition constraint; (**b**) cultivated land resource constraint; (**c**) water abundance constraint; (**d**) traffic condition constraint; and (**e**) location condition constraint types.

4.1.2. Cultivated Land Resource Constraint

Seven provinces have poverty-stricken counties that are classified under the cultivated land resource constraint type. These provinces are Jilin (0.835), Heilongjiang (0.636), Anhui (0.602), Inner Mongolia (0.569), Henan (0.496), Xinjiang (0.461), and Ningxia (0.455) (Figure 7b). The cultivated land quality determines the level of agricultural development in a county, and the agricultural economy appears to be the leading industry for most poverty-stricken counties in China. Therefore, cultivated land quality is one of the most important factors that controls the economic development of poverty-stricken counties. The hindering effects of poor cultivated land quality can accumulate over time and eventually lead to a large gap between a poverty-stricken county and other regions. Except for Henan and Anhui, the aforementioned provinces are mainly concentrated in the northern part of China, where the cultivated land in most poverty-stricken counties has poor quality. For the residents who mainly rely on the agricultural economy for their income, the poor quality of the cultivated land directly affects their living standards and becomes the main reason for their poverty.

### 4.1.3. Water Abundance Constraint

The poverty-stricken counties in Chongqing (0.643), Ningxia (0.465), Jiangxi (0.393), Hunan (0.254), Henan (0.252), Hebei (0.226), and Guangxi (0.190) are classified under the water abundance constraint type (Figure 7c). Water is vital to the development of a county's economy and the development of human society. As an important resource for agricultural irrigation, water determines the agricultural development level of a county and affects the total value of its primary industry. Moreover, water is an important resource for secondary and tertiary industries. The aforementioned provinces are concentrated in the central part of China where levels of soil erosion, desertification, and rocky desertification are high. The water scarcity in this area also restricts the development of agriculture and the construction of water-related industries, thereby limiting the economic development of its counties and leading to poverty.

### 4.1.4. Traffic Condition Constraint

The poverty-stricken counties in Ningxia (0.970), Chongqing (0.735), Heilongjiang (0.242), Qinghai (0.236), Jilin (0.204), and Henan (0.198) are classified under the traffic condition constraint type (Figure 7d). The road network density is an important indicator that reflects the conditions of transportation infrastructure in an area. The lack of transportation infrastructure restricts the economic and cultural links between a county and its outside regions and eventually leads to a vicious circle of occlusion and backwardness. The poverty-stricken counties in the aforementioned provinces are mostly located in the remote areas of the central city. The poor development of transportation infrastructure can also lead to poor accessibility of an area, thereby presenting a bottleneck for the economic development of poverty-stricken counties.

### 4.1.5. Location Condition Constraint

The poverty-stricken counties in Chongqing (0.647), Heilongjiang (0.482), Hebei (0.416), Anhui (0.326), Henan (0.325), Qinghai (0.290), Hunan (0.222), and Guangxi (0.255) are classified under the location condition constraint type (Figure 7e). The location index reflects the closeness of the economic and social links between a county and the central city and directly affects the county's economic development. A high location index corresponds to a high degree of economic and public service radiation from the central city to a county. However, the lack of economic linkages between poverty-stricken counties and advanced central cities can lead to a backward production and lifestyle for the poor population, thereby seriously limiting the economic development of their counties. The poverty-stricken counties in the aforementioned provinces are mostly located along the fringe of their respective provinces, thereby making them unable to effectively accept the advanced developmental conditions of central cities and restricting their entry to the internal economic centers of their provinces.

### 4.2. Spatial Combinations of the Formation Mechanism in Poverty-Stricken Counties

For the single-factor combination, the economic development of poverty-stricken counties is mainly affected by a single spatial influential factor (Table 9). The results of the geographical detector model identify the terrain relief and cultivated land quality as single dominant factors that shape the economic development of poverty-stricken counties. The terrain relief dominates the economic development of poverty-stricken counties in Hubei (0.181) and Shaanxi (0.198). The poverty-stricken counties with very poor and relatively poor terrain conditions account for approximately 80% of all counties in these provinces. The poverty-stricken counties whose economic development is restricted by cultivated land quality are located in Inner Mongolia (0.569) and Xinjiang (0.461). Among them, the proportion of poverty-stricken counties with the last two levels of cultivated land quality in Inner Mongolia is 42.5%, and poverty-stricken counties with very poor and relatively poor cultivated land quality account for 66.6% of all counties in Xinjiang. Therefore, the poverty-stricken counties in these

provinces should focus on the quality of their cultivated land and formulate a new development model to improve their economy and alleviate their poverty.

**Table 9.** Cluster analysis results of spatial constraints in each province.

| Cluster Type | Province | Spatial Constraint |
|---|---|---|
| Single factor | Hubei | Terrain relief 0.181 |
| | Shaanxi | Terrain relief 0.198 |
| | Inner Mongolia | Cultivated land quality 0.569 |
| | Xinjiang | Cultivated land quality 0.461 |
| Double factors | Jiangxi | Terrain relief 0.257, Water abundance 0.393 |
| | Jilin | Cultivated land quality 0.835, Road network density 0.204 |
| | Hunan | Water abundance 0.254, Location index 0.222 |
| | Guangxi | Water abundance 0.190, Location index 0.255 |
| Multiple factors | Hebei | Terrain relief 0.294, Water abundance 0.226, Location index 0.416 |
| | Heilongjiang | Cultivated land quality 0.636, Road network density 0.242, Location index 0.482 |
| | Anhui | Terrain relief 0.519, Cultivated land quality 0.602, Location index 0.326 |
| | Chongqing | Water abundance 0.643, Road network density 0.735, Location index 0.647 |
| | Henan | Terrain relief 0.363, Cultivated land quality 0.496, Water abundance 0.252, Road network density 0.198, Location index 0.325 |
| | Qinghai | Terrain relief 0.153, Road network density 0.236, Location index 0.290 |
| | Ningxia | Terrain relief 0.313, Cultivated land quality 0.455, Water abundance 0.465, Road network density 0.970 |

For the double-factor combination, the economic development of poverty-stricken counties is mainly affected by two spatial influential factors (Table 9). For instance, the topographic relief (0.257) and water abundance (0.393), cultivated land quality (0.835) and road network density (0.204), water abundance (0.254) and the location index (0.222), and water abundance (0.190) and the location index (0.255) affect the economic development of poverty-stricken counties in Jiangxi, Jilin, Hunan, and Guangxi, respectively. The above analysis confirms the feasibility of the results obtained by the geographical detector model.

In the multiple-factor combination, the economic development of poverty-stricken counties is affected by the combination of multiple spatial influential factors (Table 9). The poverty-stricken counties in Hebei, Heilongjiang, Anhui, Chongqing, and Qinghai are affected by three spatial influential factors, those in Ningxia are affected by four spatial influential factors, and those in Henan are affected by five spatial influential factors. Six other provinces have poverty-stricken counties that are influenced by the location index, while the poverty-stricken counties in five of these provinces are influenced by the terrain relief and road network density. The poverty-stricken counties with two or multiple spatial influential factors have a highly complex economic development because these multiple factors restrain one another. Therefore, the economic development of a county should be comprehensively considered along with the various aspects and should focus on economic development.

## 5. Conclusions

This paper examines the distribution pattern of the spatial influential factors of poverty-stricken counties. The geographical detector model was employed to explore the spatial formation mechanism of these counties. The regional type and main constraints of these counties are revealed, along with the internal mechanism of their spatial differentiation to provide some references that can help the government implement poverty alleviation policies.

From the perspective of the spatial distribution pattern of spatial influential factors, the poverty-stricken counties have poor natural and social conditions. This result highlights that these spatial factors significantly constrain the economic development of these counties. On the basis of the results of the geographical detector model, five types of poverty-stricken counties are classified, namely, the terrain condition constraint, the cultivated land resource constraint, the water abundance constraint,



the traffic condition constraint, and the location index constraint types. First, the poverty-stricken counties in eight provinces of China (i.e., Hubei, Shaanxi, Jiangxi, Hebei, Anhui, Henan, Qinghai, and Ningxia) are classified under the terrain condition constraint type. Second, the poverty-stricken counties in seven provinces (i.e., Inner Mongolia, Xinjiang, Jilin, Heilongjiang, Anhui, Henan, and Ningxia) are classified under the cultivated land resource constraint type. Third, the poverty-stricken counties in seven provinces (i.e., Jiangxi, Hunan, Guangxi, Hebei, Henan, Chongqing, and Ningxia) are classified under the water abundance constraint type. Fourth, the poverty-stricken counties in six provinces (i.e., Jilin, Heilongjiang, Henan, Chongqing, Qinghai, and Ningxia) are classified under the traffic condition constraint type. Finally, the poverty-stricken counties in eight provinces (i.e., Hunan, Guangxi, Hebei, Heilongjiang, Anhui, Henan, Chongqing, and Qinghai) are classified under the location condition constraint type.

In terms of the spatial combinations of their formation mechanisms, the poverty-stricken counties in four provinces (i.e., Hubei, Shaanxi, Inner Mongolia, and Xinjiang) are affected by a single-factor constraint, those in four provinces (i.e., Jiangxi, Jilin, Hunan, and Guangxi) are affected by double-factor constraints, and those in seven provinces (i.e., Hebei, Heilongjiang, Anhui, Henan, Chongqing, Qinghai, and Ningxia) are affected by multiple-factor constraints. The differentiated policies can be designed according to the spatial influential factors of poverty-stricken counties through an understanding of their spatial formation mechanisms.

**Author Contributions:** Conceptualization, L.Z. and L.X.; methodology, L.Z.; software, L.X. and Y.W.; validation, L.Z.; formal analysis, L.X.; investigation, L.Z.; writing—original draft preparation, L.Z.; writing—review and editing, F.Z. and L.X.; visualization, L.X. and Y.W.; supervision, L.Z., F.Z. and L.X.

**Funding:** This research was funded by National Natural Science Foundation of China (grant numbers 41701185 and 41601411), Jiangsu Natural Science Foundation of Youth (grant number BK20160893).

**Acknowledgments:** The authors express their gratitude towards the journal editor and the reviewers, whose thoughtful suggestions played a significant role in improving the quality of this paper.

**Conflicts of Interest:** The authors declare no conflicts of interest.

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
