# Peer review of "Modeling the Spatial Formation Mechanism of Poverty-Stricken Counties in China by Using Geographical Detector"

_sustainability, doi:10.3390/su11174752_

Round 1

Reviewer 1 Report

Comments and suggestions for the authors:

In my opinion, the manuscript is under the scope of the selected journal. However, the paper requires English language editing and there are other issues that should be addressed before the publication of the manuscript in this journal or elsewhere.

Introduction section of the paper:

1.     Lines 39-40: Please clarify that the statement is about China. Consider revising to “The imbalanced development of different regions in china has ….”

2.     Consider using more updated citations. Citation 1 refers to a publication in 1991. It is not even available in English for the readers to check. I suggest removing this citation.

3.     Lines 50-51: There is a need for citing multiple sources not just one. The authors mentioned: “many scholar…,” but the citation refers to one study, which is not a systematic review or meta-analysis. Therefore, for this statement the authors should consider citing more than one source or rephrase the statement.

4.     Line 58: Consider rephrasing the second sentence to “Wei suggested that…” to avoid using “found” in two consecutive sentences.

5.     Lines 59-60: Please add a reference/citation: “A higher poverty rate is observed in the western…recorded in the eastern and plain areas [add a citation].”

6.     Lines 63-65: Please add a reference/citation: “In addition, … alleviation efforts [add a citation].”

7.     Lines 65-67: The statement is confusing. It is not clear if  “However, the spatial….driving mechanisms” refers to the conclusion of the authors based on the discussed studies or it refers to a suggestion based on the China Rural Poverty Alleviation and Development Program? Please consider rephrasing and clarifying.  

8.     Line 69-71: “Among these factors,… social history of an area” is confusing, please consider rephrasing.

9.     Lines 78-81: Please add a reference/citation: “The places near main traffic … socio-economic development [add a citation].” Also consider changing “inaccessibility” to “lack of access to.”

10.  Lines 84-85: I am not sure why the authors mentioned “qualitative relationship?” Please consider rephrasing or delete “qualitative.”

11.  Lines 95-98: the sentence is too long, confusing, and has grammatical problem. Please consider breaking this sentence into two.

Materials and Methods section of the paper:

1.     Lines 101-103: A citation is needed for the documents that the authors referred to here.

2.     Lines 103-106: Citations are needed for the resources that the authors referred to here.

3.     Line 112: Add a citation for the 2017 China Statistical Yearbook.

4.     Lines 137-138: Consider rephrasing, I am not sure what “their influential factors” refers to?

5.     It is not clear how formula (1) was created. Please indicate if this formula has been used in other studies.

6.     Lines 166-168: Consider rephrasing the following sentence “…this percentage is far below than that recorded in the sample of poverty-stricken counties.”

7.     Lines 172-173: Consider changing “compared with” to “compared to.”

8.     Lines 217-220: Consider rephrasing: “all counties in the sample” instead of “all counties sample;” “are below the average national level” instead of “below the national level;” and “, suggesting the water abundance..,” instead of “, thereby suggesting that the water abundance.”

9.     Lines 234-237: consider deleting the first “respectively” from the sentence.

10.  Lines 237-238: please move “In sum, the road … the government” to the discussion section as this statement is more like a recommendation rather than a result.

11.  Line 242: Does “national level” refer to an average national level? If yes, please consider adding “average.”

12.  Lines 260-263: consider deleting the first “respectively” from the sentence.

Discussion section of the paper:

1.     Why Table 9 is presented in the Discussion section instead of the Result section? The relevant part should be moved to the Result section and explanation about the method of this analysis should be added to the Methods section of the paper.

2.     Consider adding some concrete policy consideration besides summarizing the results and vaguely suggesting that “differentiated policies can be designed according to the spatial influential factors of poverty-stricken counties.”

Conclusion section of the paper:

1.     Lines 418-129: Consider breaking this very long sentence into multiple sentences. Also, “those in” is confusing. Please consider referring to countries more clearly.   

Reviewer 2 Report

It is good to see the application of ‘geographical detector’ for identifying the major drivers of poverty-stricken counties in china. This work has the potential to add to existing the body of knowledge on the spatial pattern of poverty and the factors associated with it.

However, I see the need for major revision before this work is recommended for publication.  I currently see two main issues with the paper.

The first issue is with the sampling framework or lack of robust counterfactual, and the second is regarding the interpretation of the findings. Please refer to my general and pointwise comments below. 

The abstract needs to be rewritten as it assumes that the readers are familiar with certain terms that are very specific to this paper. The term “formation mechanism” should be replaced with the term more specific to the material covered in the paper.

The analysis in the paper doesn’t address the formation mechanism, rather it looks at the factors associated/present in the poverty-stricken counties.  Line 50-67: Are there any alternate findings different from what you have cited? Line 84-98: It is good to see the authors recognize spatial heterogeneity.

Also, it would be important to highlight that correlation is not causation. We might see several factors associated with the spatial patterns of poverty. However, one needs to conduct a counterfactual analysis to be able to say if there is a causality link between the factors and the patterns of poverty.  Line 113 the table heading “Accounts(%)”. Did you mean the Percentage of the total?  Figure 1: units are missing in the elevation map (b) Table 1, the indicator “Cultivated land quality”  rather represents the quantity of cultivated land normalized by population.

Similarly, the water abundance indicator assumes that there will be an equal flow of water in all the rivers. The authors have to draw on more empirical work where similar indicators are used or highlight these issues in the limitation section. 

Figure 2,3,4,5, 6  what is the type of classification used to the categorization of the factors into various classes? 

Line 308, we do not know if the counties are “greatly affected”, here again, correlation is not causation. 

The authors could consider using factor analysis to further highlight the variables associated with the poverty-stricken counties and associated factors.

Reviewer 3 Report

Broad Comments

1- While the unit of observation for this underlying work done in this paper is the county level, results are presented in the context of provinces. In section 3.1 of the results this is done as percentage of counties within each province that fall into the 5 categories. In section 3.2 of the results and in the discussion, results are presented in terms of the q values for each province. This is not inherently an issue, but the Authors never provide the reader with any context for using the provinces as aggregation. I would suggest adding a paragraph, possibly in the methods section, explaining the use of the province level in the rest of the paper. Additionally, the initial paragraph in section 3.2 seems to contradict the data that is presented in Table 8 (see final specific comment below).

2- The results from sections 3.1.x are a bit difficult to put into context as a whole. The text attempts to highlight the key points in each section, but the sometimes lengthy text-based listed of provinces and percentages do not really make it easier. One possible option to solving this issue might be adding some color highlighting to Tables 3-7 for provinces over x%, so readers can visually interpret the impact of each spatial factor across the provinces. 

Specific Comments

Section 2.1 Line 105 - Where is the DLG river network data from? Please provide a source for this in this section.

Section 2.2 Line 130 - How are the values used to split the counties into 5 categories determined?

Section 2.2 - This needs some additional detail on how the formulas for the factors in Table 2 were developed. Are these existing accepted formulas (if so, cite), or did you develop them for this paper (if so, provide some background on how).

Table 2 Water Abundance - Would it be possible to incorporate flow rate or width of river in this metric instead of length of water alone?

Line 165 - Missing space "and10.3%"

Figure 2-6 (c) - I would suggest adding x and y axis labels

Section 3.1.x - In each of these sections I would spend a sentence or two stating what the "good" and "poor" value range is and how that was split across the 5 categories.

Section 3.1.x - The first paragraph of each of these sections (line 171, 194, 216, 240, 267) has a sentence such as "... increased by 21%, 5.3%, -8%..." and I would suggest changing it to "...changed by +21%, +5.3%, -8%..."

Line 217 - States that "relatively few counties in the poverty-stricken counties sample have relatively good levels of water abundance" but Figure 4c shows that the percentages for all counties vs poverty-stricken counties is nearly identical.

Section 3.2 Line 284 - This states that q values are assessed at county level vs average of counties in a province, yet all subsequent results seem to be comparing province level vs average of provinces in country. (See Broad Comment #1)

Reviewer 4 Report

This study assessed the spatial relationship of county distribution with five spatial influential factors, terrain relief, cultivated land quality, water resource abundance, road network density, location index. I think it has been well organized, with clear statement of the question, adequate description of the methods and results.

Q1: It is not clear to me that how are the poverty-stricken counties classified as very poor, relatively poor, fair, relatively good, and good?

Q2: The factor, terrain relief, seems not that representative. Other topographic value, such as slope, might be better.

Reviewer 5 Report

The paper is good for the publication in the present form

Author Response

Response: We appreciate the reviewer’s positive comment of our paper. In addition, we have made some necessary revisions according to the suggestions from other reviewers.

Round 2

Reviewer 2 Report

The manuscript reads much better, and the authors have addressed most of the technical issues raised by me. I  would recommend it for publication after proofreading for language issues.